# LPS-induced systemic inflammation is suppressed by the PDZ motif peptide of ZO-1 *via* regulation of macrophage M1/M2 polarization

**Hyun-Chae Lee[1†], Sun-Hee Park[1†], Hye Min Jeong[1], Goeun Shin[2], Sung In Lim[2], Jeongtae Kim[3], Jaewon Shim[4], Yeong-Min Park[5]\*, Kyoung Seob Song[1]\***

[1]Department of Medical Science, Kosin University College of Medicine, Busan, Republic of Korea; [2]Department of Chemical Engineering, Pukyong National University, Busan, Republic of Korea; [3]Department of Anatomy, Kosin University College of Medicine, Busan, Republic of Korea; [4]Department of Biochemistry, Kosin University College of Medicine, Busan, Republic of Korea; [5]Department of Integrative Biological Sciences and Industry, College of Life Sciences, Sejong University, Seoul, Republic of Korea

**\*For correspondence:**
immun3023@sejong.ac.kr (Y-MinP);
kssong@kosin.ac.kr (KSeobS)

[†]These authors contributed equally to this work

**Competing interest:** The authors declare that no competing interests exist.

**Abstract** The gram-negative bacterium lipopolysaccharide (LPS) is frequently administered to generate models of systemic inflammation. However, there are several side effects and no effective treatment for LPS-induced systemic inflammation. PEGylated PDZ peptide based on zonula occludens-1 (ZO-1) was analyzed for its effects on systemic inflammation induced by LPS. PDZ peptide administration led to the restoration of tissue injuries (kidney, liver, and lung) and prevented alterations in biochemical plasma markers. The production of pro-inflammatory cytokines was significantly decreased in the plasma and lung BALF in the PDZ-administered mice. Flow cytometry analysis revealed the PDZ peptide significantly inhibited inflammation, mainly by decreasing the population of M1 macrophages, and neutrophils (immature and mature), and increasing M2 macrophages. Using RNA sequencing analysis, the expression levels of the NF-κB-related proteins were lower in PDZ-treated cells than in LPS-treated cells. In addition, wild-type PDZ peptide significantly increased mitochondrial membrane integrity and decreased LPS-induced mitochondria fission. Interestingly, PDZ peptide dramatically could reduce LPS-induced NF-κB signaling, ROS production, and the expression of M1 macrophage marker proteins, but increased the expression of M2 macrophage marker proteins. These results indicated that PEGylated PDZ peptide inhibits LPS-induced systemic inflammation, reducing tissue injuries and reestablishing homeostasis, and may be a therapeutic candidate against systemic inflammation.

## eLife assessment

This **important** study identifies the anti-inflammatory function of PEGylated PDZ peptides that are derived from the ZO-1 protein. Results from cellular and in vivo experiments tracking key inflammatory markers are **compelling**. Although the present study would benefit from investigating chronic inflammation conditions using microbe and protein data, the work provides a proof of concept for developing novel strategies against acute inflammatory conditions such as sepsis.

## Introduction

Systemic inflammation causes significant morbidity and mortality worldwide (*Juskewitch et al., 2012*) and is a common cause of pathogenesis for multiple degenerative diseases, such as cardiovascular disease, cancer, diabetes mellitus, chronic kidney disease, non-alcoholic fatty liver disease, auto-immune and neurodegenerative disorders, and coronary heart disease (*Qin et al., 2007*; *Furman et al., 2019*; *Sattar et al., 2003*). One major source of morbidities is pro-inflammatory cytokines from immune cells and activation of the innate immune system against pathogens (*Furman et al., 2019*). However, the physiological mechanisms and therapeutic candidates against LPS-induced systemic inflammation are poorly understood.

ZO-1 is a tight junction protein that regulates cell permeability in many mammalian cells and functions as a scaffolding protein. ZO-1 consists of three PDZ domains, a SH3 domain, and a guanylate kinase (GuK) domain. Because the ZO-1 protein strictly and completely regulates cell permeability, ZO-1 overexpression could inhibit LPS- and PM2.5-induced respiratory inflammation by diminishing the production of pro-inflammatory cytokines (*Lee et al., 2020*; *Kang et al., 2020*). Notably, among ZO-1 proteins, PDZ could participate in PDZ-PDZ interactions and bind with non-PDZ-containing proteins (*Lee and Zheng, 2010*). The PDZ peptide consists of a core sequence that significantly decreases LPS-induced F-actin polymerization and production of pro-inflammatory cytokines and/or chemokines to maintain homeostasis (*Lee et al., 2020*). In addition, RGS12 expression activated by the PDZ peptide can completely block LPS-induced CXCR2 GPCR activation (*Lee et al., 2020*). Although the PDZ peptide can regulate LPS-and PM2.5-related airway inflammation, this is limited to the respiratory system. The effects of the PDZ peptide on LPS-induced systemic inflammation are poorly understood.

M1 and M2 macrophage polarization differs in specific functions in the immune system (*Zhang et al., 2023*; *Li et al., 2023*). M1 macrophages are the initial immune response to infections and can produce pro-inflammatory cytokines. M2 macrophages are mediated for anti-inflammation and tissue repair and have specific physiological functions to produce anti-inflammatory cytokines and tissue remodeling (*Li et al., 2023*). Polarization of macrophages into these subtypes is important because it determines the progression to inflammation or anti-inflammation. Neutrophils are a type of white blood cell that mainly assist the body in fighting off infections (*Chung et al., 2023*). Immature neutrophils can be present but do not participate in infection control, while mature neutrophils are fully functional and primed for fighting infections (*Chung et al., 2023*). Therefore, the presence of immature neutrophils in the bloodstream may be an indication of inflammation or infection.

Here, we suggest that PEGylated PDZ peptide inhibits LPS-induced systemic inflammation by hindering ROS production in the kidney mitochondria. Because the PDZ peptide restored LPS-induced tissue injury in the liver, kidney, and lung in mice, it likely inhibits the major symptoms of systemic inflammation and may be an innovative therapeutic candidate for sepsis and septic shock.

## Results
### ZO-1-based PDZ peptide is generated and PEGylated

Based on our previous studies (*Lee et al., 2020*; *Kang et al., 2020*), ZO-1 protein potentially regulates LPS- or PM2.5-induced airway inflammation by enforcing membrane tightness. In addition, the PDZ domain plays an essential role in LPS- or PM2.5-induced ZO-1 physiological function by generating TEER. Therefore, we hypothesized that PDZ peptide downregulates systemic LPS inflammation. The PDZ peptide consisting of 22 amino acids and an additional TAT (GRKKRRQRRR) sequence to enable cell permeability was tagged with FITC for monitoring (*Figure 1a*). The clinical application of a therapeutic peptide is often limited by its short in vivo half-life. PEGylation is one of the well-established approaches for extending in vivo half-life of a peptide drug; A PEGylated peptide is expected to be more tolerable to renal clearance and proteolytic degradation. To PEGylate WT PDZ or mut PDZ at its amine groups, m-PEG-succinimidyl succinate with an average molecular weight of 5000 Da was incubated with WT PDZ or mut PDZ to yield mono- and di-PEGylated WT PDZ or mut PDZ, collectively referred to as PEG-WT-PDZ or PEG-mut PDZ, respectively. Size exclusion chromatography on the PEGylation mixture resulted in the separation of PEG-WT-PDZ or PEG-mut PDZ from unreacted PEG and residual WT PDZ or mut PDZ (*Figure 1b*). PEG-WT PDZ or PEG-mut PDZ collected from eluting fractions was analyzed by in-gel fluorescence upon SDS-PAGE (*Figure 1c*).

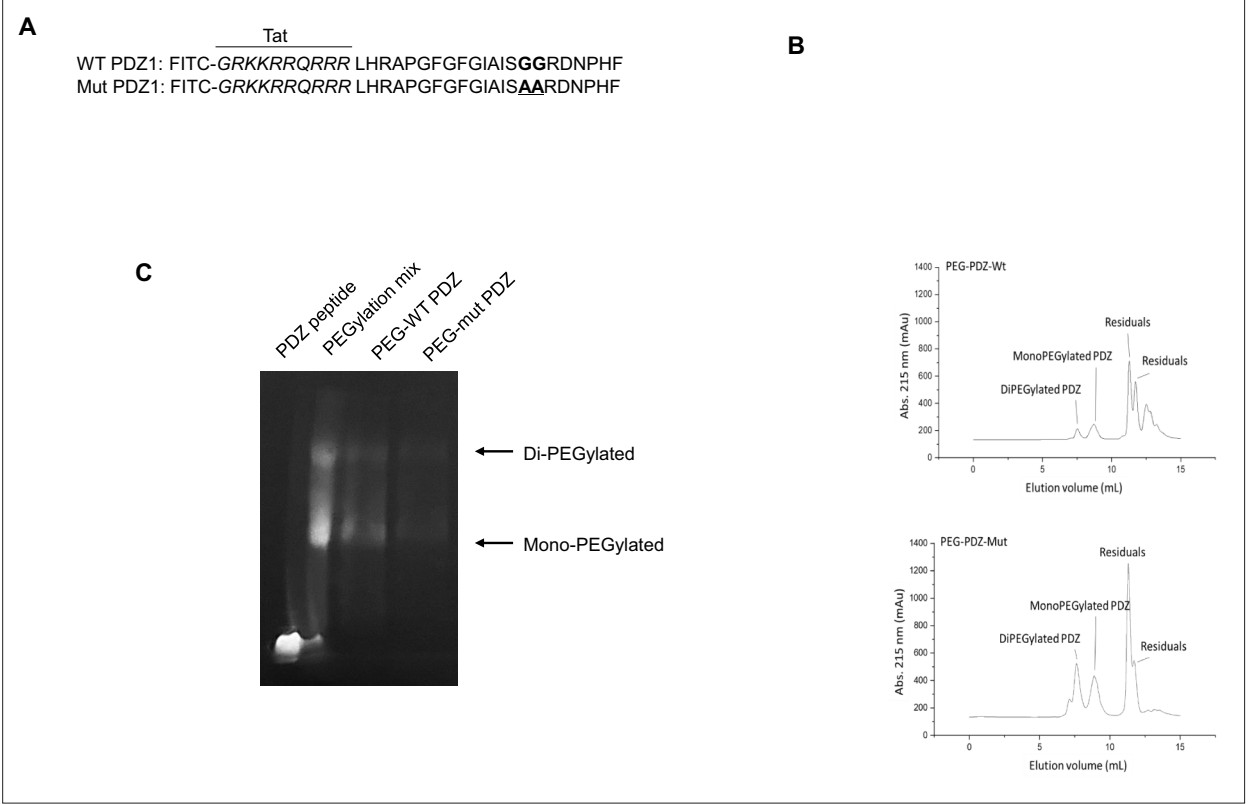

**Figure 1.** PEGylation and purification of PEGylated PDZ (PEG-WT PDZ) and PEGylated mut PDZ (PEG-mut PDZ). (**A**) Peptides were synthesized with fluorescein isothiocyanate (FITC) and entered the Tat region (italicized amino acids) for cell penetration based on the first PDZ domain sequence (A, upper panel), and 25GG26 was changed to 25AA26 for the mutant PDZ peptide (mut PDZ peptide; lower panel). (**B**) Size exclusion chromatography for isolation of PEG-WT PDZ or PEG-mut PDZ from the mixture of WT PDZ or mut PDZ reacted with m-PEG-succinimidyl succinate (MW 5000). (**C**) SDS-PAGE analysis for visualization of PEGylated products. Samples were run on SDS-PAGE and then illuminated at $\lambda$ ex = 480 nm to obtain in-gel fluorescence.

The online version of this article includes the following source data for figure 1:

**Source data 1.** Labelled gel for **Figure 1C**.

**Source data 2.** Raw unedited gels for **Figure 1C**.

In comparison to the PEGylation mixture which contained native PDZ as well as PEGylated PDZs, purified PEG-WT PDZ or PEG-mut PDZ was comprised of only mono- and di-PEGylates with fairly retarded migration. These results indicated that PEGylation was successfully achieved through amine-coupling at lysine residues of PDZs and substantially increased the hydrodynamic volume of PDZs.

**Table 1.** The effects of PDZ peptide on lipopolysaccharide (LPS)-induced damage markers in plasma.

Five days after LPS administration (10 mg/kg/30 µL) into the peritoneum of mice that were injected with either the wild-type (WT) PDZ peptide or the mutant PDZ peptide (mPDZ peptide; 7.5 mg/kg/30 µL, i.v.) 4 hr previously. The blood samples were centrifuged, and the plasma was harvested. The ALT, AST, BUN, and creatinine levels were analyzed (n=5).

| Plasma maker | Control | LPS only | LPS+WT PDZ | LPS+mut PDZ |
|---|---|---|---|---|
| ALT (U/L) | 8.286±2.456 | 11.785±4.167 | 8.024±0.24 | 6.16±1.917 |
| AST (U/L) | 54.555±14.702 | 82.306±20.569 | 62.176±14.275 | 65.445±16.042 |
| BUN (mg/dl) | 5.716±1.646 | 7.742±2.317 | 9.120±0.215 | 11.996±1.51 |
| Creatinine (mg/dl) | 0.172±0.095 | 0.472±0.065 | 0.214±0.048 | 0.434±0.078 |

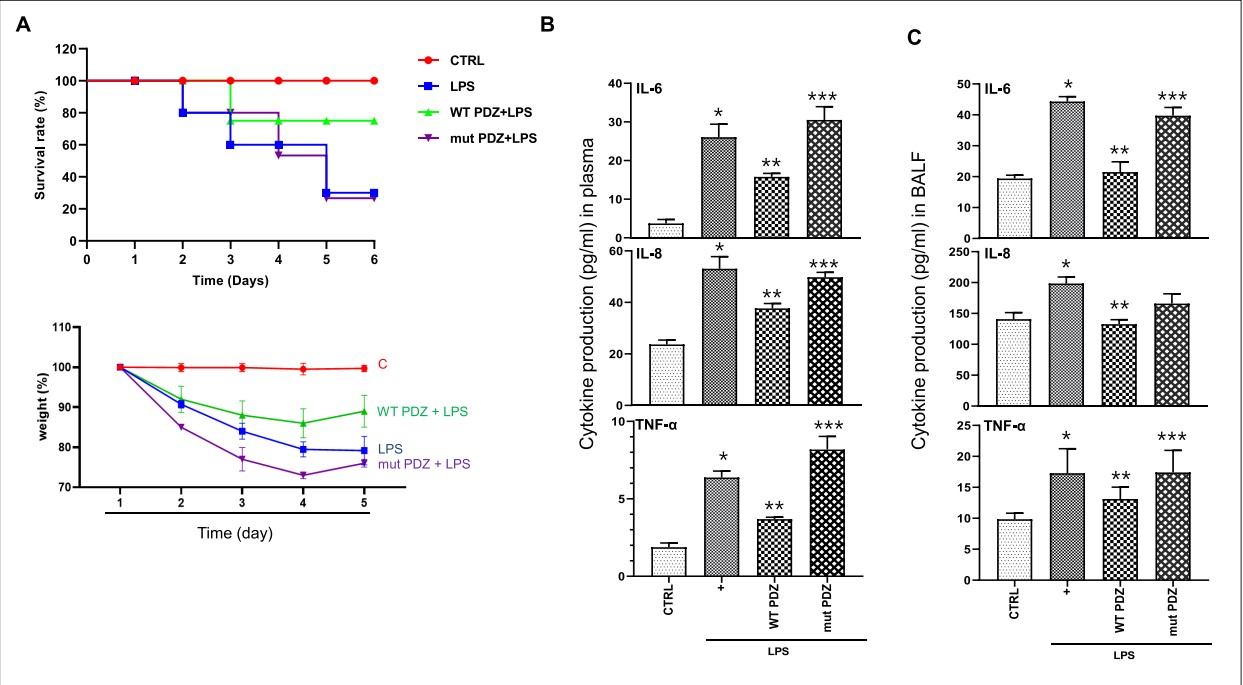

**Figure 2.** The inhibitory effect of PDZ peptide on lipopolysaccharide (LPS)-induced systemic inflammatory responses in vivo. (**A**) Comparison of survival rate and body weight change after administration of LPS or both LPS and PDZ peptide. The body weight was measured for the duration of LPS administration (5 days). LPS administration (10 mg/kg/30 µL) into the peritoneum of mice that were injected 4 hr prior with either the wild-type (WT) PDZ peptide or the mutant PDZ peptide (7.5 mg/kg/30 µL, i.v.). (**B**) After 5 days, the IL-6, IL-8, and TNF-α concentrations in the plasma were measured using specific ELISAs. (**C**) To examine lung pathology, BALF was harvested. The IL-6, IL-8, and TNF-α concentrations in the BALF were measured using specific ELISAs. *$p < 0.05$ compared with saline-treated mice; **$p < 0.05$ compared with LPS-treated mice; ***$p < 0.05$ compared with LPS- and WT PDZ peptide-treated mice.

## PDZ peptide exerts anti-inflammatory effects against LPS-induced systemic inflammation in LPS-administered mice

We examined the anti-inflammatory effects of the PDZ peptide on LPS-administered mice. Mice were pretreated with either WT or mut PDZ peptide (7.5 mg/kg, i.v.) for 4 hr and then injected with LPS (10 mg/kg, i.p.) to induce experimental systemic inflammation. After sacrifice, blood samples were collected, and the ALT, AST, BUN, and creatinine levels were measured to identify the changes in plasma biochemical markers (*Table 1*). A significant increase in plasma biomarkers of systemic inflammation was observed in LPS-administered mice. Notably, WT PDZ-treated mice had decreased levels of biomarkers compared to LPS-treated mice. However, mut PDZ-treated mice had not changed, except creatinine, suggesting that WT PDZ peptide significantly decreased LPS-induced elevation of creatinine in plasma. Although the function of ZO-1 protein as a diagnostic marker for sepsis was identified in 2020 year (*Ni et al., 2020*), additional studies have not been performed. Here, the in vivo survival effect of LPS and PDZ co-administration was examined in mice. The pretreatment with WT PDZ peptide significantly increased survival and rescued compared to LPS only; these effects were not observed with the mut PDZ peptide (*Figure 2a*). To investigate whether WT PDZ peptide can affect pro-inflammatory cytokine production/recruitment in the plasma after LPS administration, various pre-inflammatory cytokines were measured after blood perfusion through the heart. The IL-6, IL-8, and TNF-α levels were significantly decreased in the plasma of WT PDZ peptide-administered mice, but not in mut PDZ peptide-administered mice (*Figure 2b*). Next, systemic inflammation can cause septic shock and severe damage to lung, kidney, liver, and other organs (*Caraballo and Jaimes, 2019*; *Lelubre and Vincent, 2018*). Thus, lung inflammation was examined under the same conditions. The levels of pro-inflammatory cytokines were examined using specific ELISAs (*Figure 2c*). The LPS-induced pro-inflammatory cytokines production was significantly inhibited in WT PDZ peptide-administered mice, but not in mut PDZ peptide-administered mice. These results indicate that the WT

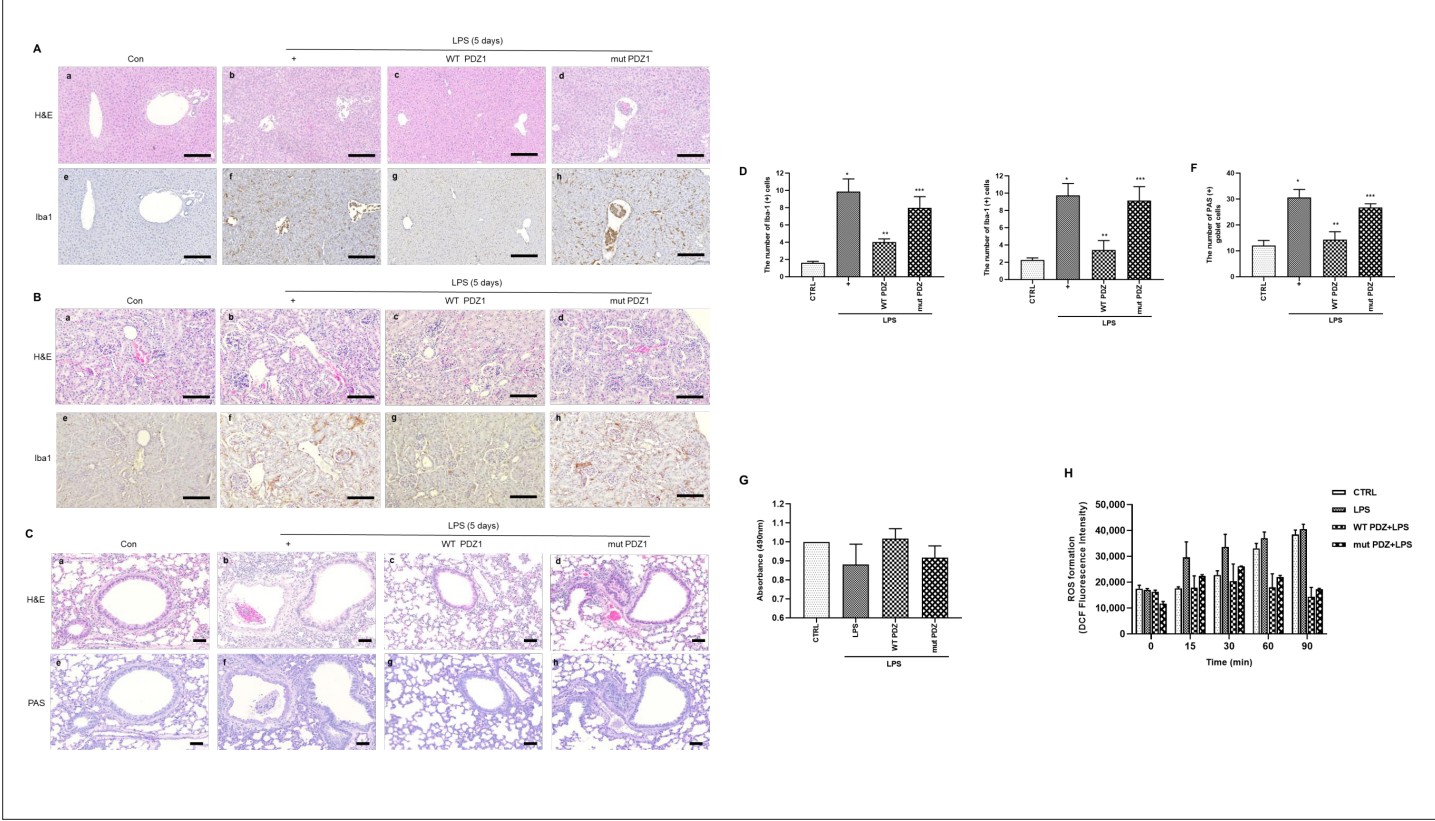

**Figure 3.** The effects of PDZ peptide on liver, kidney, and lung injuries in lipopolysaccharide (LPS)-administered mice. LPS administration (10 mg/kg/30 μL) into the peritoneum of mice injected 4 hr prior with either the wild-type (WT) PDZ peptide or the mutant PDZ peptide (7.5 mg/kg/30 μL, i.v.). After administration for 5 days, the liver (a), kidney (b), and lung (c) were harvested. Both hematoxylin and eosin (H&E) and iba-1 staining were performed on the tissue sections from the treated mice (400x). Indicated scale bars were 20 μm. (d,e,f) Semi-quantitative analysis of each is positive for Iba-1 in the liver and kidney, and positive cells of periodic acid Schiff (PAS) in the lung, respectively. (g) After the kidneys were harvested, tissue lysates were used for MTT assay. (h) After mitochondria from the kidneys was isolated, the mitochondria lysates were used for ROS measurement. *p<0.05 compared with saline-treated mice; **p<0.05 compared with LPS-treated mice; ***p<0.05 compared with LPS- and WT PDZ peptide-treated mice. All data shown are representative of three independent experiments.

PDZ peptide significantly inhibits the LPS-induced the loss of weight and pro-inflammatory cytokine production to maintain homeostasis in blood and lungs after LPS administration.

## PDZ inhibits LPS-induced tissue injury by blocking ROS formation

Next, we assessed whether WT PDZ peptide affects acute tissue damage in organs such as the liver, kidney, and lung during LPS-induced systemic inflammation. The liver, kidney, and lung samples were collected and histopathologically examined. LPS administration significantly enhanced inflammatory cell infiltration in the liver (*Figure 3a*), kidney (*Figure 3b*), and lung (*Figure 3c*) tissue samples. Interestingly, these infiltrations were significantly inhibited by WT PDZ administration, but not mut PDZ in the liver, kidney, and lung. And the immunohistochemistry staining using macrophage and microglia marker, Iba1, is helpful for the indication of an inflamed site. Iba1 staining was strongly positive in LPS-administrated Liver and kidney. Interestingly, this iba1 staining was dramatically decreased in WT PDZ treatment, but not mut PDZ in the liver and kidney. In addition, LPS administration increased PAS-positive goblet cells in lung tissue (*Figure 3c*). PAS-positive goblet cells were strongly decreased by WT PDZ administration, but not by mut PDZ administration. The specific physiological mechanism by which WT PDZ peptide decreases LPS-induced systemic inflammation in mice and the signal molecules involved remain unclear. These were confirmed by a semi-quantitative analysis of Iba-1 immunoreactivity and PAS staining in the liver, kidney, and lung, respectively (*Figure 4d, e and f*). To examine whether WT PDZ peptide can alter LPS-induced tissue damage in the kidney, a cell toxicity assay was performed (*Figure 3g*). LPS induced cell damage in the kidney, however, WT PDZ peptide could

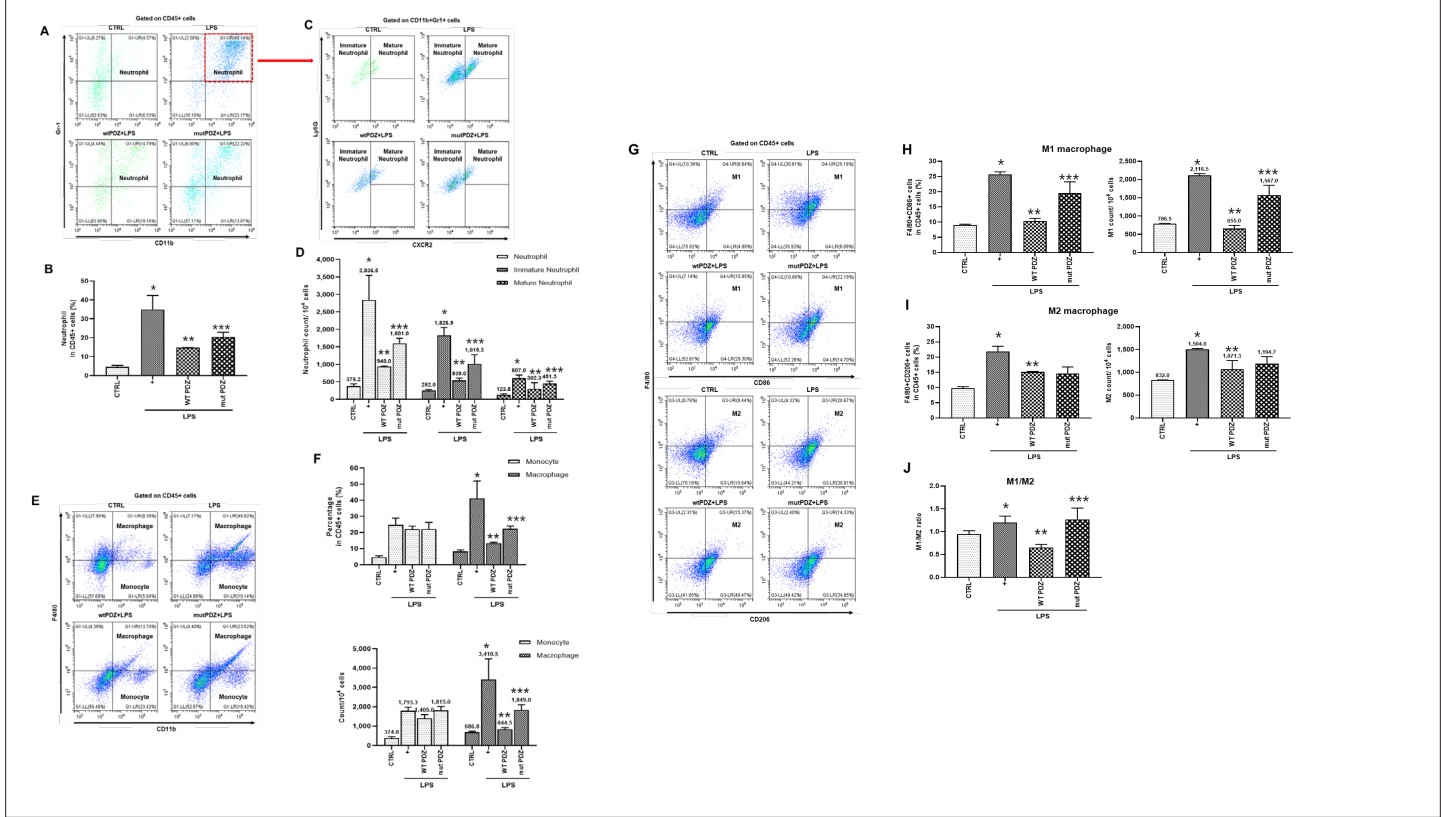

**Figure 4.** Gating strategy used in flow cytometry to determine the populations of immune cells in mice administered lipopolysaccharide (LPS) and PDZ peptides. Five days after LPS administration (10 mg/kg/30 μL) into the peritoneum of mice that were injected 4 hr prior with either the wild-type (WT) PDZ peptide or the mutant PDZ peptide (7.5 mg/kg/30 μL, i.v.). After administration for 5 days, spleens were harvested. (**A, C**) Subpopulations of either immature or mature neutrophils were identified as CXCR2-Ly6Glo/+ (immature) and CXCR2+Ly6G+ (mature). (**B, D**) Data for each neutrophil phenotype are displayed as the percentages of neutrophil (CD11b+Gr-1+) in CD45+ cells and the exact number of immature- and mature-neutrophil counted per 104 total neutrophil cells. (**E, F**) Flow cytometry of monocyte (CD11b+F4/80int) and macrophage (CD11b+F4/80+) expression in CD45+ cells is shown. Data for each cell phenotype are displayed as the percentages of monocytes and macrophages in CD45+ cells and the exact number of monocytes and macrophages counted per 104 total single cells. (**G**) Subpopulations of two types of macrophages were identified as CD86+F4/80+ (M1) and CD206+F4/80+ (M2) in CD45+ cells. (**H, I**) Data for each cell phenotype are displayed as the percentages of M1- and M2-macrophages in CD45+ cells and the exact number of M1- and M2- macrophages counted per 104 total single cells. (**J**) The M1/M2 ratio was calculated based on the percentage of F4/80+CD86+ cells and F4/80+CD206+ cells in CD45+ cells. *p<0.05 compared with saline-treated mice; **p<0.05 compared with LPS-treated mice; ***p<0.05 compared with LPS- and WT PDZ peptide-treated mice. All data shown are representative of three independent experiments.

significantly alleviate the toxicity, but mut PDZ peptide could not. Because cytotoxicity caused by LPS is frequently due to ROS production in the kidney (*Su et al., 2023*; *Qiongyue et al., 2022*), ROS production in the mitochondria was investigated in renal mitochondria cells harvested from kidney tissue (*Figure 3h*). LPS significantly increased ROS production to induce renal inflammation in a time-dependent manner; however, the WT PDZ peptide blocked LPS-induced ROS production, but the mut PDZ peptide did not. The results indicate that the WT PDZ peptide may have therapeutic effects that inhibit LPS-induced tissue damage and systemic inflammation in several organs of mice administered LPS.

## PDZ peptide regulates LPS-induced immune populations by inhibiting M1 macrophage polarization and promoting M2 polarization

We examined whether PDZ peptide can affect the population of immune cells. Screening of immune cells was performed for immature/mature neutrophils, monocytes, and macrophages in the spleen from the mice by flow cytometry. Although proliferative immature/mature neutrophils were examined after treatment with LPS, the effect of PDZ peptide on an LPS-induced neutrophil lineage was unclear. To investigate this phenomenon, the neutrophil fraction was analyzed with CD11b⁺Gr1⁺ in the spleen

(*Figure 4a*). LPS significantly increased the percentage of total neutrophils. The WT PDZ peptide robustly decreased total neutrophils, however, the mut PDZ peptide increased compared to WT PDZ-treated mice (*Figure 4a and b*). Analyzing total neutrophils in the spleen, there were two populations of neutrophils: a population of CXCR2 +Ly6G+ neutrophils (Mature) and a second population of CXCR2-Ly6Glo/+ neutrophils (immature) (*Figure 4c*). Both immature and mature neutrophils were dramatically increased by LPS. Importantly, the WT PDZ peptide robustly decreased both immature and mature neutrophils, however, mut PDZ peptide increased them compared to WT PDZ-treated mice (*Figure 4d*). Next, LPS markedly increased the percentage of CD11b+F4/80int monocytes. There were no differences between the WT PDZ- and mut PDZ-treated mice in the spleen. However, the CD11b+F4/80+ macrophages were dramatically increased in the LPS-treated mice (*Figure 4e and f*). The WT PDZ peptide robustly decreased macrophages, however, the mut PDZ peptide increased compared to WT PDZ-treated mice (*Figure 4e and f*). Lastly, to examine whether the PDZ peptide could regulate the M1 and M2 macrophages population and polarization in vivo, spleen cells were stained to characterize the cell populations (*Figure 4g*). The F480 +CD86+ M1 macrophages were dramatically increased in the LPS-treated mice. The WT PDZ peptide remarkably decreased M1 macrophages, however, mut PDZ peptide increased compared to WT PDZ-treated mice (*Figure 4h*). The percentage of F4/80+CD206+M2 macrophages were also increased in the LPS-treated mice, however, there were no significant differences between the WT PDZ- and mut PDZ-treated mice (*Figure 4i*). More importantly, there were no significant differences of the M1/M2 ratio between the LPS- and mut PDZ-treated mice, whereas the ratio was dramatically decreased in WT PDZ-treated mice (*Figure 4j*). The decrease in the M1/M2 ratio meant that the M2 was relatively increased, suggesting that WT PDZ peptide can inhibit M1 macrophage polarization and promotes M2 macrophage polarization to decrease LPS-induced systemic inflammation. These results show that WT PDZ peptide may inhibit LPS-induced tissue damage and systemic inflammation by regulating the population of immune cells to keep homeostasis during LPS-induced systemic inflammation.

## NF-κB- mediated gene expression levels are downregulated in PDZ peptide-treated cells

Although the physiological function of PDZ peptide has become increasingly important in systemic inflammation, the signaling pathway has not yet been fully elucidated. To identify PDZ-mediated signaling, we performed RNA sequencing analysis. The results of RNA-seq analysis showed the expression pattern of 24,424 genes according to each comparison combination, of which the results showed the similarity of 51 genes overlapping in four gene categories and the similarity between each comparison combination (*Figure 5a*). As a result, compared to the control group, it was confirmed that LPS alone, WT PDZ+LPS, and mut PDZ+LPS were all upregulated above the average value in each gene, and when LPS treatment alone was compared with WT PDZ+LPS, it was confirmed that they were averaged or downregulated. When comparing LPS treatment alone and mut PDZ+LPS, it was confirmed that about half of the genes were upregulated. Regarding the similarity between comparison combinations, the comparison combination with LPS/Control and the comparison combination with WT PDZ+LPS/Control showed the most similar gene patterns, followed by the comparison combination with mut PDZ+LPS/Control. The comparison combination of WT PDZ+LPS/LPS and mut PDZ+LPS/LPS showed similar genetic patterns. Similarity between genes showed that inflammation-related genes had increased expression in the LPS/Control, WT PDZ+LPS/Control, and mut PDZ+LPS/Control treatment groups. NF-kB signaling pathway appeared most frequently in WT PDZ+LPS/LPS (91.46%), Inflammatory autoimmunity appeared most frequently in mut PDZ+LPS/LPS (47.75%), and inflammatory cytokines receptors and inflammasomes also appeared in mut PDZ+LPS/LPS (*Figure 5b*). It appeared the most at 33.33% and 61.42%. In addition, increased and decreased expression of genes in the inflammatory cytokines receptors gene category was observed in LPS/Control, WT PDZ+LPS/LPS, and mut PDZ+LPS/LPS. In addition, the STRING database was used to represent protein-protein interactions of genes in RNA-seq analysis (*Figure 5c*). In the network shown, genes served as nodes and edges served as protein-protein associations. The line thickness at the network edge indicated the strength of data support. Interaction scores were considered to have minimal reliability at 0.400. Focusing on NF-kB, mTOR, REL, SAA1, and CXCL3 had strong protein-protein interactions. Therefore, we thought the NF-kB signaling would be dominated for the anti-inflammatory signaling of PDZ peptide.

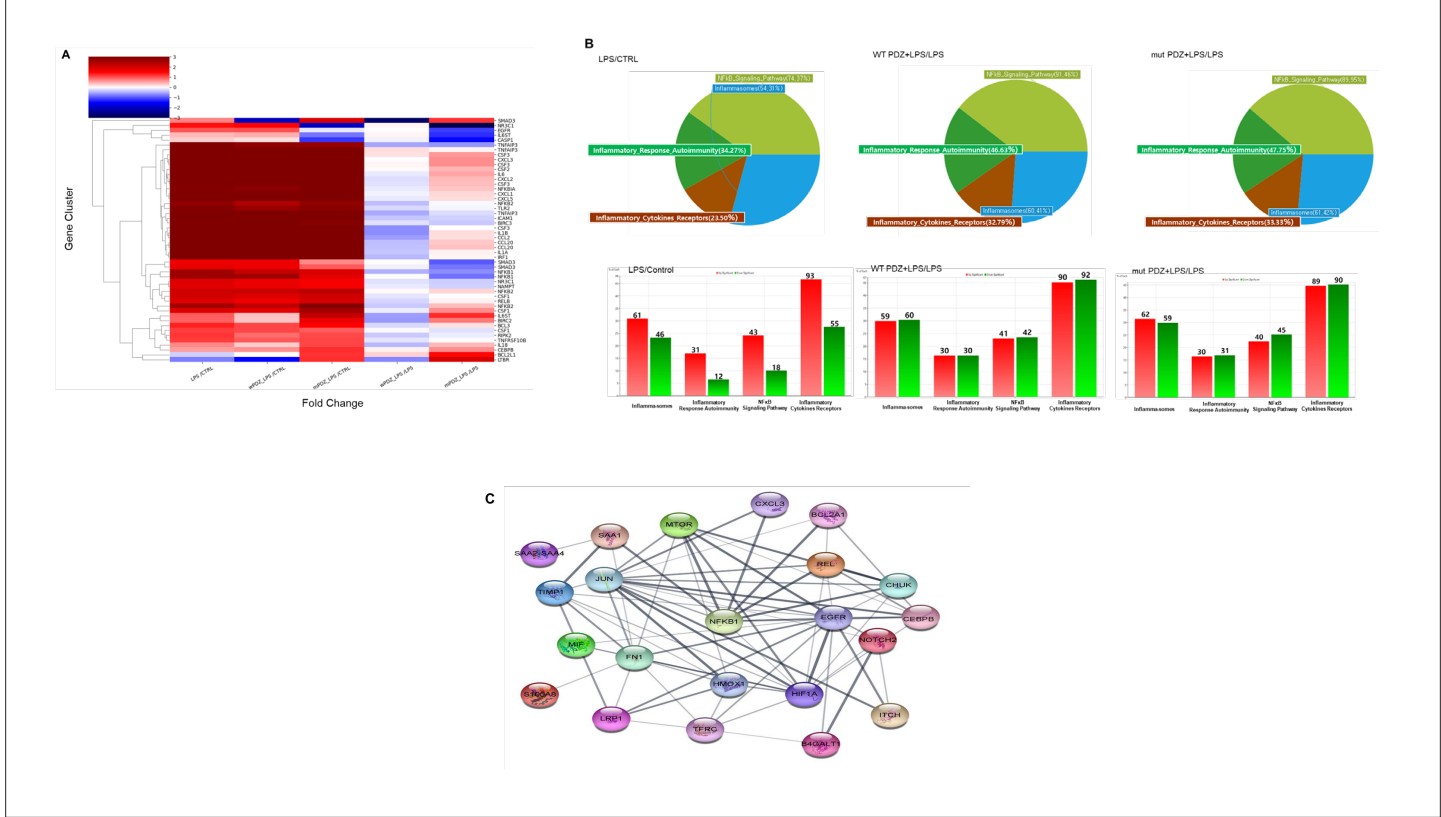

**Figure 5.** Comparison of mRNA expression in PDZ-treated BEAS-2B cells. BEAS-2B cells were treated with wild-type PDZ or mutant PDZ peptide for 24 hr and then incubated with lipopolysaccharide (LPS) for 2 hr, after which RNA sequencing analysis was performed. (**A**) The heat map shows the general regulation pattern of about 51 inflammation-related genes that are differentially expressed when WT PDZ and mut PDZ are treated with LPS, an inflammatory substance. All samples are RED = upregulated and BLUE = downregulated relative to the gene average. Each row represents a gene, and the columns represent the values of the control group treated only with LPS and the WT PDZ and mut PDZ groups with LPS. This was used by converting each log value into a fold change value. All genes were adjusted to have the same mean and standard deviation, the unit of change is the standard deviation from the mean, and the color value range of each row is the same. (**B**) Significant genes were selected using Gene category chat (Fold change value of 2.00 and normalized data (log2) value of 4.00). The above pie chart shows the distribution of four gene categories when comparing LPS versus control, WT PDZ +LPS/LPS, and mut PDZ +LPS/LPS. The bar graph below shows RED = upregulated, and GREEN = downregulated for each gene category, and shows the number of upregulated and downregulated genes in each gene category. (**C**) The protein-protein interaction network constructed by the STRING database differentially displays commonly occurring genes by comparing WT PDZ +LPS/LPS, mut PDZ +LPS/LPS, and LPS. These nodes represent proteins associated with inflammation, and these connecting lines denote interactions between two proteins. Different line thicknesses indicate types of evidence used in predicting the associations.

## PDZ peptide restores LPS-induced mitochondrial dynamics via NF-kB signaling pathway and ROS production in Raw264.7 cells

To examine which PDZ peptide affects LPS-induced mitochondrial membrane integrity in Raw 264.7 cells, mitotracker 488 mitochondria dye was utilized. The mitoTracker staining was dramatically decreased in a LPS-time dependent manner (*Figure 6a*). Interestingly, wild-type PDZ peptide significantly increased the staining compared to LPS only (*Figure 6b*). These findings suggest that PDZ peptide restored the LPS-induced mitochondrial membrane integrity. However, the mutant PDZ peptide did not have an effect. In addition, the phospho-Drp1 antibody was used to detect LPS-induced mitochondria fission in Raw 264.7 cells. The phospho-Drp1 staining was increased in an LPS-time-dependent manner (*Figure 6c*). Interestingly, wild-type PDZ peptide significantly decreased the staining compared to LPS only (*Figure 6d*), suggesting that PDZ peptide can regulate mitochondria fission by controlling phosphorylation of Drp1 to decrease LPS-induced inflammation in Raw264.7 cells. Based on the results of RNA-sequencing analysis (*Figure 5*), to investigate which signal transduction pathway is induced in the cells stimulated by LPS, LPS was treated in a time-dependent manner. The Western blot analysis was performed using several phospho-specific antibodies (*Figure 6e*).

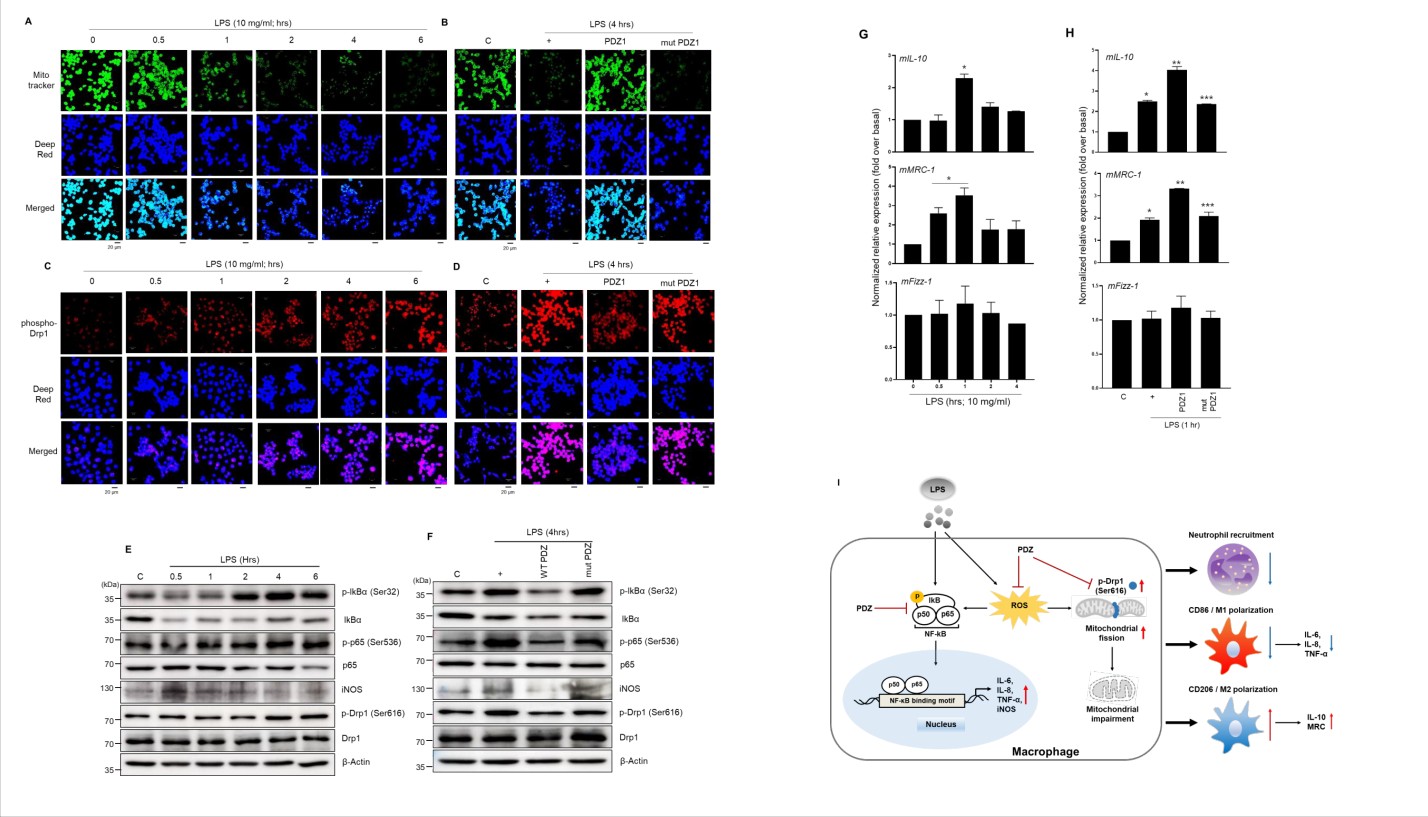

**Figure 6.** PDZ peptide dramatically increases M2 marker proteins to regulate M1/M2 polarization via NF-$\kappa$B signaling and ROS production. (**a, c, e, g**) The cells were treated with lipopolysaccharide (LPS) in a time-dependent manner. After treatment with wild-type (WT) PDZ and mutant PDZ, the cells were treated with LPS for 4 hr (**b, d, f**) or 1 hr (**h**). The mitochondria morphology was stained using MitoTracker Green FM and visualized (**a, b**). The mitochondria fission was stained using phospho-Drp1 antibody and visualized (**c, d**).Indicated scale bars were 20 µm. The phospho-specific and total antibodies were assessed by Western blot analysis. β-actin was used as a loading control (**e, f**). The lysates were prepared for qPCR for the expression of M1 and M2 marker proteins (**g, h**). *p<0.05 compared with control; **p<0.05 compared with LPS treatment; ***p<0.05 compared with WT PDZ-treated transfectants. All of the data shown are representative of three independent experiments. (**i**) A graphic abstract illustrating that the PDZ peptide of zonula occludens-1 (ZO-1) protein inhibits LPS-induced systemic inflammation by regulating the activation of NF-$\kappa$B signaling pathways, ROS production, and M1/M2 polarization.

The online version of this article includes the following source data for figure 6:

**Source data 1.** Uncropped and labelled gels for *Figure 6E, F*.

**Source data 2.** Raw unedited gels for *Figure 6E, F*.

LPS-induced the phosphorylation of iκB and p65 was significantly peaked at 4 hr and the expression of iNOS by LPS reached a maximum at 30 min and decreased. However, no change in total κB and p65 expressions was observed. Phosphorylation of Drp1 was performed to observe it at the protein level. The phosphorylation of Drp1 by LPS reached a maximum at 4 hr. Interestingly, wild-type PDZ peptide significantly inhibited the expression of iNOS and the phosphorylation of iκB, p65, and Drp1, but not mutant PDZ peptide (*Figure 6f*), suggesting that NF-κB signaling and ROS production appear to be closely related during PDZ-restored mitochondria dynamics. In addition, we investigated whether wild-type PDZ peptides can alter the expression of specific marker proteins for M1 and M2 polarization. Wild-type PDZ peptide significantly decreased the expression of M1 marker protein (mIL-6, mIL-8, and mTNF-α) (*Figure 2b*). In addition, the expression of M2 marker protein (mIL-10 and mMRC-1) were increased at 1 hr after the treatment of LPS, but not mFizz-1 (*Figure 6g*). As opposed to M1 marker proteins, the expression of M2 marker proteins (mIL-10 and mMRC-1) was significantly increased in PDZ-treated cells compared to LPS only, but not mut PDZ peptide (*Figure 6h*). These results confirmed the observations of the flow cytometry assay performed above (*Figure 4h and i*). These results show that PDZ peptide dramatically regulates mitochondrial membrane integrity and mitochondrial dynamics via NF-κB signaling and ROS production.

## Discussion

Systemic inflammation can cause many clinical diseases with high mortality and sepsis-associated organ dysfunction (*Dolin et al., 2019*). In the present study, we hypothesized that a tight junction protein could affect systemic inflammatory response in LPS-administered mice. The main results are as follows: PDZ peptide administration into LPS-challenged mice significantly decreased pro-inflammatory cytokine production in plasma and BALF and diminished oxidative stress in the kidney; PDZ peptide downregulated immune cell population in spleen cells. Thus, the data indicate the PEGylated PDZ peptide as a therapeutic candidate for LPS-induced acute systemic inflammation.

The ZO-1 protein includes protein-protein interaction domains (three PDZ domains) and signaling domains (*Hung and Sheng, 2002*). In addition, the ZO-1 protein consists of SH3 and catalytically inactive GuK domains. In our previous studies (*Lee et al., 2020*; *Kang et al., 2020*), the PDZ peptide strongly inhibited LPS- or PM2.5-induced airway inflammation in the mice respiratory tract. The physiological functions identified were as follows: PDZ peptide strongly suppressed LPS- or PM2.5-induced F-actin polymerization to result in decreased airway inflammation. Also, expression of CXCR2 as the receptor for IL-8 secretion activated by LPS was significantly inhibited by PDZ, whereas RGS12 expression that has a GTPase-activating function increased after PDZ peptide treatment and downregulated CXCR2 receptor activation. However, these phenomena were confined to only the respiratory tract. We could not determine whether ZO-1 is involved in LPS-induced systemic inflammation or if the PDZ peptide could downregulate systemic inflammation. Therefore, we are currently investigating the physiological symptoms of systemic inflammation caused by LPS in *ZO-1* conditional knock-out mice.

In the present study, the PEGylated PDZ peptide was considered an innovative therapeutic candidate to decrease LPS-induced systemic inflammation in mice. However, although colistin has poor efficacy and renal toxicity, it is a powerful antibiotic used as a last resort to treat infections caused by multidrug-resistant Gram-negative bacteria (*Sabnis et al., 2021*). In addition, treatment failure occurs frequently, and the efficacy of colistin is very low (*Paul et al., 2018*; *MacNair et al., 2018*; *Satlin et al., 2020*). Furthermore, colistin has significant renal and neurological side effects that generate oxidative stress (*Spapen et al., 2011*). However, the PDZ peptide in the present study did not have any side-effects and decreased ROS generation in renal mitochondria (*Figure 3*). Nevertheless, the physiological mechanisms by which the PDZ peptide can control LPS-induced systemic inflammation in mice and the signal molecules involved, especially in the downstream signaling of G-protein, remain unclear. Hypothetically, because PDZ has a TAT sequence as a cell-penetrating peptide (CPP), the peptide easily enters the mitochondria to regulate ROS generation. The PDZ peptide inhibited iNOS and eNOS activities in human bronchial epithelial cells (BEAS-2B; data not shown), but the mut PDZ peptide did not. In our previous studies (*Lee et al., 2020*; *Kang et al., 2020*; *Kim and Song, 2022*), LPS secreted many cytokines and/or chemokines extracellularly to signaling pathways associated with inflammation to nearby cells, inducing severe cytokine storm in several organs. However, the PDZ peptide strongly inhibited these extracellular secretions at the cellular level. Because the PDZ domain can bind to many signaling-related proteins through PDZ-PDZ interactions (*Lee and Zheng, 2010*; *Lee et al., 2006*; *Gisler et al., 2008*), the signaling complex might regulate systemic inflammation. To investigate PDZ-binding partners in mammalian cells, the GST:PDZ complex was immunoprecipitated from BEAS-2B cells transfected with the construct. The binding proteins are currently being identified using tandem mass spectrometry analysis.

Neutrophils dominate the initial stage of inflammation by pathogen invasion, and macrophages are activated and differentiated by receiving signals from the microenvironment of damaged tissues, which are coordinated by the expression of numerous cytokines and receptors (*Butterfield et al., 2006*). Therefore, neutrophils and macrophages perform various roles, such as the defense of the body from pathogens, regulation of the inflammatory environment, tissue development, homeostasis, etc., and are therapeutic targets related to a wide range of inflammation. The spleen functions as a unique extramedullary reservoir for myeloid cells, including monocytes, macrophages, and neutrophils. Recently, Deniset et al. revealed the existence of two distinct neutrophil populations within the red pulp (RP) of the spleen: the mature Ly6G$^{high}$ neutrophils, and the immature Ly6G$^{int}$ neutrophils (*Deniset et al., 2017*). Therefore, in the present study, the relationship between PDZ peptide and two distinct neutrophil populations was investigated in the spleen. The percentage and number of total neutrophils dramatically increased by LPS treatment were significantly decreased by WT PDZ peptide, but mut PDZ peptide increased them compared to the WT PDZ peptide. These results, also,

were the same in the analysis of immature and mature neutrophils. Interestingly, the proportion of immature neutrophils was much higher than that of mature neutrophils during LPS-induced systemic inflammation (*Figure 4*). These were similar to the results found by Deniset et al. That is, after infection with *Streptococcus pneumonia*, Ly6G^high mature neutrophils within the RP of the spleen were replenished by Ly6G^int immature neutrophils. This phenomenon was thought to be due to the local increase of Ly6G^int immature neutrophils as a consequence of an extramedullary hematopoiesis and their differentiation into Ly6G^high mature neutrophils (*Deniset et al., 2017*). Next, we investigated the changes in monocytes and macrophages by PDZ peptide during LPS-induced systemic inflammation. Although the WT PDZ peptide did not reduce the increased monocytes caused by LPS-induced inflammation, it dramatically decreased the percentage and number of macrophages (*Figure 4*). In particular, the WT PDZ peptide dramatically reduced the M1 macrophages increased by LPS and significantly decreased the M1/M2 ratio, thereby promoting the M2 polarization of macrophages (*Figure 4*). The results indicated that the proinflammatory microenvironment of the LPS challenge might increase M1 macrophage polarization by inflammatory cytokines and/or chemokines inhibiting M2 macrophage polarization. However, the WT PDZ peptide treatment significantly inhibited M1 polarization to activate M2 polarization. Therefore, the results of the present study demonstrate an innovative therapeutic candidate for an increased ratio of M2 macrophages in the immune cell population during LPS-induced systemic inflammation. In addition, the PDZ peptide can alter the immature/mature neutrophil ratio and population of immune cells. These results indicate that the PDZ peptide may up-regulate the production of anti-inflammatory cytokines and/or chemokines such as IL-4, TGF-β, IL-13, IL-10, and M-CSF to polarize M2 macrophages and the population of immune cells (*Ahmed and Ismail, 2020*; *Kong et al., 2015*). Furthermore, if the mechanism of M1/M2 polarization of macrophages and recovery of inflammatory tissue could be identified using PDZ peptide in vivo, it may apply to the study of various diseases caused by macrophages.

Recently, mitochondria have an important role in the control of inflammatory responses (*Marchi et al., 2023*). In addition, mitochondria have emerged as being necessary for both the establishment and maintenance of innate and adaptive immune cell responses (*Sandhir et al., 2017*; *Weinberg et al., 2015*). Thus, we focused on the key function of mitochondria in regulating the innate immune response and the mitochondrial dysfunction known to be promoted by inflammation, we investigated whether mitochondria that modulate inflammatory signaling are regulated by PDZ peptides. LPS-induced mitochondrial membrane integrity and fission are critical in the inflammatory response, and we showed that PDZ regulates LPS-induced systemic inflammation by modulating these phenomena. These results can be explained as follows: (1) PDZ peptide inhibited LPS-induced NF-κB signal pathway (*Figure 6f*). While it is already known that NF-κB signaling plays an important role in mitochondria dynamics (*Laforge et al., 2016*), this is the first study to show that inhibition of NF-kB signaling by PDZ affected the mitochondrial dynamics. (2) ROS production by iNOS also was suppressed by the PDZ peptide (*Figure 6f*). ROS induced by potassium deprivation increased the phosphorylation of Drp1 to mitochondrial fragmentation (*Cid-Castro and Morán, 2021*). Although more precise studies are needed, the inhibition of iNOS expression by PDZ peptide inhibited ROS formation (*Figures 3e and 6f*). (3) The phosphorylation of Drp1 has been known to be critical for mitochondria fission (*Xie et al., 2020*). The phosphorylation of Drp1 was significantly decreased by PDZ peptide in LPS-treated cells (*Figure 6d and f*). These results strongly suggest that the function of PDZ peptide can regulate the mitochondrial dynamics, and is a very important phenomenon that suppresses LPS-induced systemic inflammation through the mitochondria dynamics (*Figure 6i*).

In summary, two analyses of organ injury and spleen cells were used in the present study to understand the relationship between tissue injury and immune cells. We hypothesized that treatment with PDZ peptide at the early stage of LPS-induced infection would downregulate the inflammatory progress by activating homeostatic tight junction proteins and anti-inflammatory responses. Therefore, the PDZ peptide may be a therapeutic candidate for LPS-induced systemic inflammation including that of human pulmonary diseases.

# Materials and methods

## Peptide PEGylation

Both wild-type (WT) PDZ and mutant PDZ (mut PDZ) peptides were synthesized by Peptron (Daejeon, Korea). WT PDZ or mut PDZ in 1 mM of PBS at pH 7.4 was mixed with a fivefold molar excess of mPEG-NHS (MW 5000) at a concentration of 0.5–2.0 mg/mL and incubated at room temperature (RT) for 1 hr. Size exclusion chromatography was performed on the ÄKTA PRIME chromatographic system to isolate PEGylated WT PDZ or mut PDZ (PDZ-PEG or mut PDZ-PEG) using Superdex 200 increase 10/300 GL column. The reaction mixture was applied to the column pre-equilibrated with PBS and eluted at 0.02 CV/min with detection at 215 nm. Fractions were analyzed using SDS-PAGE in a tricine buffer system or appropriately concentrated or diluted for further evaluation.

## Mice and instillation

Ten-week-old male C57BL/6 mice were maintained in accordance with the guidelines and under the approval of the Animal Care Committee of Kosin University College of Medicine, Busan, Korea (KMAP-23–18). Mice were randomly divided into four groups (n=5/group) and challenged with WT PDZ or mut PDZ peptide (7.5 mg/kg, i.v.) prior to LPS injection (10 mg/kg, i.p.).

## Tissue collection and plasma biomarker biochemistry

Mice were anesthetized with both Alfaxan (80 mg/kg, i.p.) and Rompun (10 mg/kg, i.p.), and their blood and tissues (kidney, liver, and lung) were collected. Blood was obtained from the vena cava into heparin/EDTA-coated tubes and centrifuged at 4000×g for 30 min at RT. Plasma biochemistry analysis (ALT, AGT, creatinine, and BUN) was performed by GC Labs (Yong-in, Korea).

## Tissue preparation and histological examination

Tissues were fixed with 4% paraformaldehyde in phosphate buffer (100 mM, pH 7.4). Samples were embedded in paraffin wax, and 5 µm-thickness sections were prepared using a rotary microtome (Leica RM 2135, Leica, Nussloch, Germany). Paraffin sections were deparaffinized in xylene and dehydrated in serial ethanol solutions, followed by hematoxylin and eosin (H&E), Periodic acid Schiff (PAS) staining, and immunohistochemistry staining.

## Immunohistochemistry

Immunohistochemistry was performed using the Vectastain Elite ABC Kit (Vector Laboratories). Briefly, deparaffinized sections were treated with citrate buffer (10 mM, pH 6.0) in a microwave for 3 min, and then treated with 0.3% hydrogen peroxide in methyl alcohol for 20 min to block endogenous peroxidase activity. Subsequently, sections were incubated with the matching blocking serum (10% [v/v] goat serum in phosphate buffer saline (PBS); Vectastain Elite ABC Kit; Vector Laboratories). The samples were incubated with primary antibody, rabbit anti-ionized calcium binding adaptor molecule 1 (Iba1; 1:1000, 019–19749, LER0547, Wako) for 1 hr at room temperature. After three washes with PBS, sections were incubated with biotinylated rabbit IgG antibody and then with ABC peroxidase, according to the manufacturer's instructions. The peroxidase reaction was developed using a 3–3'-diaminobenzidine substrate kit (Vector Laboratories), followed by counterstaining with hematoxylin before mounting.

## Renal mitochondria isolation, MTT assay, and ROS measurement

Isolation of kidney mitochondria was performed using a specific assay kit (Thermo Fisher Scientific) according to the manufacturer's instructions. The CellTiter 96 AQ$_{ueous}$ One solution cell proliferation assay kit (Promega) was used for cell toxicity. Isolated mitochondria samples (200 mg) were homogenized in 2 mL of ice-cooled Tris-Cl buffer (40 mM, pH 7.4, 4 °C). Then, 100 µL of tissue homogenate was mixed with 1 mL of Tris-Cl buffer (40 mM, pH 7.4) and 10 µM of 2',7'dichlorofluorescein diacetate. The mixture was incubated in the dark for 15 min at 37 °C. Finally, the fluorescence intensity of the samples was measured using a FLUOstar Omega (BMG Labtech, Offenburg, Germany) multifunctional microplate reader ($\lambda$ excitation: 485 nm and $\lambda$ emission: 525 nm). Fluorescence was measured at four-time points post-incubation (0, 30, 60, and 90 min) (*Heidari et al., 2019*).

## Tissue preparation and flow cytometry

Spleens were harvested and homogenized into single-cell suspensions using a biomasher tube (Kimble Chase) and 70 μm polypropylene nylon mesh. Collected cells were lysed in red blood cell lysis buffer (Gibco, Thermo Fisher Scientific) and washed twice with PBS at 1500×g for 5 min at 4 °C. Cell pellets were resuspended in FACS buffer containing 0.1% BSA and 1% FBS. Antibodies were purchased from BioLegend. For identification of mouse myeloid cells, cells were stained with fluorochrome-conjugated anti-mouse antibodies against CD45 (FITC, clone 30-F11), CD11b (PE, clone M1/70), F4/80 (APC/Cyanine7, clone BM8), Gr1 (APC, clone RB6-8C5), Ly6G (PerCP/Cyanine5.5, clone 1A8), CXCR2 (APC/Cyanine7, clone SA044G4), CD86 (PerCP/Cyanine5.5, clone GL-1), or CD206 (APC, clone C068C2). Appropriate isotype control antibodies were used when necessary. After staining, labeled cells were washed twice with FACS buffer at 1500×g for 5 min at 4 °C and analyzed using CytoFLEX (Beckman Coulter Diagnostics, La Brea, CA, USA).

## Library preparation and sequencing

Libraries were prepared from total RNA using the NEBNext Ultra II Directional RNA-Seq Kit (NEW ENGLAND BioLabs, Inc, UK). The isolation of mRNA was performed using the Poly(A) RNA Selection Kit (LEXOGEN, Inc, Austria). The isolated mRNAs were used for the cDNA synthesis and shearing, following the manufacture's instructions. Indexing was performed using the Illumina indexes 1–12. The enrichment step was carried out using of PCR. Subsequently, libraries were checked using the TapeStation HS D1000 Screen Tape (Agilent Technologies, Amstelveen, Netherlands) to evaluate the mean fragment size. Quantification was performed using the library quantification kit using a StepOne Real-Time PCR System (Life Technologies, Inc, USA). High-throughput sequencing was performed as paired-end 100 sequencing using NovaSeq 6000 (Illumina, Inc, USA).

## Data analysis

A quality control of raw sequencing data was performed using FastQC. Adapter and low-quality reads (<Q20) were removed using FASTX_Trimmer and BBMap. Then the trimmed reads were mapped to the reference genome using TopHat. The RC (Read Count) data were processed based on the FPKM + Geometric normalization method using EdgeR within R (*R Development Core Team, 2020*). FPKM (Fragments Per kb per Million reads) values were estimated using Cufflinks. Data mining and graphic visualization were performed using ExDEGA (Ebiogen Inc, Korea). DEGs were mapped to the search tool for retrieval of interacting genes (STRING) to acquire protein-protein interaction networks (Cytoscape, http://cytoscape.org).

## Analyses and measurements of mitochondrial morphology

Raw 264.7 cells were cultured on a coverslip inside a culture dish filled with the media. When the cells reached the desired confluency, the media was removed and 500 nM of MitoTracker Green FM (Thermo Fisher Scientific) were added to visualize the mitochondrial network. The mixture was incubated for 15–45 min at 37 °C. After the staining was completed, the staining solution was replaced with fresh media, and the mitochondrial morphology was observed using fluorescence microscopy.

## Western blot

We purchased antibodies against phospho-IκBα (Ser32, # 2859), IκBα (#9242), phospho-NF-κB p65 (Ser536, #3033), NF-κB p65 (#8242), and β-actin (#8457) from Cell Signaling Technology; iNOS (ab79342) from Abcam; DRP1 (sc-271583) from Santa Cruz; Phospho-Drp1 (PA5-64821) form Thermo Fisher. Anti-mouse IgG HRP (#7076) and anti-rabbit IgG (#7074) HRP were purchased from Cell Signaling Technology as secondary antibodies. The cells were lysed with RIPA lysis buffer (50 mM Tris pH 7.4, 250 mM NaCl, 1% NP40, 0.05% SDS, 2 mM EDTA, 0.5% Deoxycholic acid, 10 mM β-glycerol phosphate, 5 mM NaF, 1 mM Na3VO4, protease inhibitor cocktail). Equal amounts of whole cell lysates were resolved by SDS-PAGE and transferred to the PVDF membrane. The membranes were blocked with 5% BSA for 1 hr at room temperature. Blots were then incubated overnight with specific antibodies in TTBS (0.5% Tween 20 in Tris-buffered saline), respectively. After rinsing with TTBS, the blots were further incubated for 45 min at room temperature with a secondary antibody in TTBS and visualized using the ECL system.*Figure 6.*

## Acknowledgements

This work was supported by a National Research Foundation of Korea (NRF) grant funded by the Korean government (MSIT) (2023R1A2C3002483 to both YMP and KSS and NRF-2021R1A4A1031380 to JK and KSS).

## Additional information

### Funding

| Funder | Grant reference number | Author |
|---|---|---|
| National Research Foundation of Korea | 2023R1A2C3002483 | Yeong-Min Park Kyoung Seob Song |
| National Research Foundation of Korea | 2021R1A4A1031380 | Jeongtae Kim Kyoung Seob Song |

The funders had no role in study design, data collection and interpretation, or the decision to submit the work for publication.

### Author contributions

Hyun-Chae Lee, Sun-Hee Park, Yeong-Min Park, Kyoung Seob Song, Conceptualization, Investigation, Writing - original draft, Writing - review and editing; Hye Min Jeong, Data curation, Validation, Investigation; Goeun Shin, Resources, Investigation; Sung In Lim, Resources, Supervision; Jeongtae Kim, Jaewon Shim, Supervision, Investigation

### Author ORCIDs

Sung In Lim ⓘ https://orcid.org/0000-0003-0564-7893
Kyoung Seob Song ⓘ https://orcid.org/0000-0003-1343-2163

### Ethics

Ten-week-old male C57BL/6 mice were maintained in accordance with the guidelines and under approval of the Animal Care Committee of Kosin University College of Medicine, Busan, Korea (KMAP-23-18).

Reviewer #2 (Public review): https://doi.org/10.7554/eLife.95285.4.sa1
Author response https://doi.org/10.7554/eLife.95285.4.sa2

## Additional files

### Supplementary files

• MDAR checklist

### Data availability

Sequencing data have been deposited and publicly available in Gene Expression Omnibus (GEO) under accession number GSE275635. https://www.ncbi.nlm.nih.gov/geo/query/acc.cgi?acc=GSE275635. All data generated or analysed during this study are included in the manuscript and supporting files; source data files have been provided for Figure 6.

The following dataset was generated:

| Author(s) | Year | Dataset title | Dataset URL | Database and Identifier |
|---|---|---|---|---|
| Song KS | 2024 | LPS-induced systemic inflammation is suppressed by the PDZ motif peptide of ZO-1 via regulation of macrophage M1/M2 polarization | https://www.ncbi.nlm.nih.gov/geo/query/acc.cgi?acc=GSE275635 | NCBI Gene Expression Omnibus, GSE275635 |

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
