## [Editor Report · eLife assessment]

This **important** study identifies the anti-inflammatory function of PEGylated PDZ peptides that are derived from the ZO-1 protein. Results from cellular and in vivo experiments tracking key inflammatory markers are **compelling**. Although the present study would benefit from investigating chronic inflammation conditions using microbe and protein data, the work provides a proof of concept for developing novel strategies against acute inflammatory conditions such as sepsis.

---

## [Referee Report · Reviewer #2 (Public review)]

Summary:

The authors investigated systemic inflammation induced by LPS in various tissues and also examined immune cells of the mice using tight junction protein-based PDZ peptide. They explored the mechanism of anti-systemic inflammatory action of PDZ peptides, which enhanced M1/M2 polarization and induced the proliferation of M2 macrophages. Additionally, they insisted the physiological mechanism that inhibited the production of ROS in mitochondria, thereby preventing systemic inflammation.

Strengths:

In the absence of specific treatments for septic shock or sepsis, the study demonstrating that tight junction-based PDZ peptides inhibit systemic inflammation caused by LPS is highly commendable. Whereas previous research focused on antibiotics, this study proves that modifying parts of intracellular proteins can significantly suppress symptoms caused by septic shock. The authors expanded the study of localized inflammation caused by LPS or PM2.5 in the respiratory tract to systemic inflammation, presenting promising results. They not only elucidated the physiological mechanism by identifying the transcriptome through RNA sequencing but also demonstrated that PDZ peptides inhibit the production of ROS in mitochondria and prevent mitochondrial fission. This research is highly regarded as an excellent study with potential as a treatment for septic shock or sepsis.

Weaknesses:

(1) They Focused intensively on acute inflammation for a short duration instead of chronic inflammation.

(2) LPS was used to induce septic shock but administrating actual microbes such as *E. coli* would yield more accurate results.

(3) The authors used pegylated peptides, but future research should utilize the optimized peptides to derive the optimal peptide, and further, PK/PD studies are also necessary.

---

## [Author Response]

The following is the authors’ response to the previous reviews.

**Reviewer #1:**
(1) Peptides were synthesized with fluorescein isothiocyanate (FITC) and Tat tag, and then PEGylated with methoxy PEG Succinimidyl Succinate.I have two concerns about the peptide design. First, FTIC was intended "for monitoring" (line 129), but was never used in the manuscript. Second, PEGylation targets the two lysine sidechains on the Tat, which would alter its penetration property.

(1) We conducted an analysis of the cellular trafficking of FITC-tagged peptides following their permeabilization into cells.

**Author response image 1. sa2fig1:** 

However, we did not include it in the main text because it is a basic result.

(2) As can be seen in the figure above, after pegylation and permeabilization, the cells were stained with FITC. It appears that this does not affect the ability to penetrate into the cells.

(2) "Superdex 200 increase 10/300 GL column" (line 437) was used to isolate mono/di PEGylated PDZ and separate them from the residual PEG and PDZ peptide. "m-PEG-succinimidyl succinate with an average molecular weight of 5000 Da" (lines 133 and 134).To my knowledge, the Superdex 200 increase 10/300 GL column is not suitable and is unlikely to produce traces shown in Figure 1B.

As Superdex 200 increase 10/300 GL featrues a fractionation range of 10,000 to 600,000 Da, we used it to fractionate PEGylated products including DiPEGylated PDZ (approx. 15 kDa) and MonoPEGylated PDZ (approx. 10 kDa) from residuals (PDZ and PEG), demonstrating successful isolation of PEGylated products (Figure 1C). Considering the molecular weights of PDZ and PEG are approximately 4.1 kDa and 5.0 kDa, respectively, the late eluting peaks from SEC were likely to represent a mixed absorbance of PDZ and PEG at 215 nm.

However, as the reviewer pointed out, it could be unreasonable to annotate peaks representing PDZ and PEG, respectively, from mixed absorbance detected in a region (11-12 min) beyond the fractionation range.

In our revised manuscript, therefore, multiple peaks in the late eluting volume (11-12 min) were labeled as 'Residuals' all together. As a reference, the revised figure 1B includes a chromatogram of pure PDZ-WT under the same analytic condition.

Therefore, we changed Fig.1B to new results.

(3) "the in vivo survival effect of LPS and PDZ co-administration was examined in mice. The pretreatment with WT PDZ peptide significantly increased survival and rescued compared to LPS only; these effects were not observed with the mut PDZ peptide (Figure 2a)." (lines 159-160).Fig 2a is the weight curve only. The data is missing in the manuscript.

We added the survived curve into Fig. 2A.

(4) Table 1, peptide treatment on ALT and AST appears minor.

In mice treated with LPS, levels of ALT and AGT in the blood are elevated, but these levels decrease upon treatment with WT PDZ. However, the use of mut PDZ does not result in significant changes. Figure 3A shows inflammatory cells within the central vein, yet no substantial hepatotoxicity is observed during the 5-day treatment with LPS. Normally, the ranges of ALT and AGT in C57BL6 mice are 16 ~ 200 U/L and 46 ~ 221 U/L, respectively, according to UCLA Diagnostic Labs. Therefore, the values in all experiments fall within these normal ranges. In summary, a 5-day treatment with LPS induces inflammation in the liver but is too short a duration to induce hepatotoxicity, resulting in lower values.

(5) MitoTraker Green FM shouldn't produce red images in Figure 6.

We changed new results (GREEN one) into Figs 6A and B.

(6) Figure 5. Comparison of mRNA expression in PDZ-treated BEAS-2B cells. Needs a clearer and more detailed description both in the main text and figure legend. The current version is very hard to read.

We changed Fig. 5A to new one to understand much easier and added more detailed results and figure legend.

Results Section in Figure 5:

we performed RNA sequencing analysis. The results of RNA-seq analysis showed the expression pattern of 24,424 genes according to each comparison combination, of which the results showed the similarity of 51 genes overlapping in 4 gene categories and the similarity between each comparison combination (Figure 5a). As a result, compared to the control group, it was confirmed that LPS alone, WT PDZ+LPS, and mut PDZ+LPS were all upregulated above the average value in each gene, and when LPS treatment alone was compared with WT PDZ+LPS, it was confirmed that they were averaged or downregulated. When comparing LPS treatment alone and mut PDZ+LPS, it was confirmed that about half of the genes were upregulated. Regarding the similarity between comparison combinations, the comparison combination with LPS…

Figure 5 Legend Section:

Figure 5. Comparison of mRNA expression in PDZ-treated BEAS-2B cells.

BEAS-2B cells were treated with wild-type PDZ or mutant PDZ peptide for 24 h and then incubated with LPS for 2 h, after which RNA sequencing analysis was performed. (a) The heat map shows the general regulation pattern of about 51 inflammation-related genes that are differentially expressed when WT PDZ and mut PDZ are treated with LPS, an inflammatory substance. All samples are RED = upregulated and BLUE = downregulated relative to the gene average. Each row represents a gene, and the columns represent the values of the control group treated only with LPS and the WT PDZ and mut PDZ groups with LPS. This was used by converting each log value into a fold change value. All genes were adjusted to have the same mean and standard deviation, the unit of change is the standard deviation from the mean, and the color value range of each row is the same. (b) Significant genes were selected using Gene category chat (Fold change value of 2.00 and normalized data (log2) value of 4.00). The above pie chart shows the distribution of four gene categories when comparing LPS versus control, WT PDZ+LPS/LPS, and mut PDZ+LPS/LPS. The bar graph below shows RED=upregulated, GREEN=downregulated for each gene category, and shows the number of upregulated and downregulated genes in each gene category. (c) The protein-protein interaction network constructed by the STRING database differentially displays commonly occurring genes by comparing WT PDZ+LPS/LPS, mut PDZ+LPS/LPS, and LPS. These nodes represent proteins associated with inflammation, and these connecting lines denote interactions between two proteins. Different line thicknesses indicate types of evidence used in predicting the associations.

**Reviewer #2:**
(1) In this paper, the authors demonstrated the anti-inflammatory effect of PDZ peptide by inhibition of NF-kB signaling. Are there any results on the PDZ peptide-binding proteins (directly or indirectly) that can regulate LPS-induced inflammatory signaling pathway? Elucidation of the PDZ peptide-its binding partner protein and regulatory mechanisms will strengthen the author's hypothesis about the anti-inflammatory effects of PDZ peptide.

As mentioned in the Discussion section, we believe it is crucial to identify proteins that directly interact with PDZ and regulate it. This direct interaction can modulate intracellular signaling pathways, so we plan to express GST-PDZ and induce binding with cellular lysates, then characterize it using the LC-Mass/Mass method. We intend to further research these findings and submit them for publication.

(2) The authors presented interesting insights into the therapeutic role of the PDZ motif peptide of ZO-1. PDZ domains are protein-protein interaction modules found in a variety of species. It has been thought that many cellular and biological functions, especially those involving signal transduction complexes, are affected by PDZ-mediated interactions. What is the rationale for selecting the core sequence that regulates inflammation among the PDZ motifs of ZO-1 shown in Figure 1A?

The rationale for selecting the core sequence that regulates inflammation among the PDZ motifs of ZO-1, as shown in Figure 1A, is grounded in the specific roles these motifs play in signal transduction pathways that are crucial for inflammatory processes. PDZ domains are recognized for their ability to function as scaffolding proteins that organize signal transduction complexes, crucial for modulating cellular and biological functions. The chosen core sequence is particularly important because it is conserved across ZO-1, ZO-2, and ZO-3, indicating a fundamental role in maintaining cellular integrity and signaling pathways. This conservation suggests that the sequence’s involvement in inflammatory regulation is not only significant in ZO-1 but also reflects a broader biological function across the ZO family.

(3) In Figure 3, the authors showed the representative images of IHC, please add the quantification analysis of Iba1 expression and PAS-positive cells using Image J or other software. To help understand the figure, an indication is needed to distinguish specifically stained cells (for example, a dotted line or an arrow).

We added the semi-quantitative results into Figs. 3d,e,f.

Result section: The specific physiological mechanism by which WT PDZ peptide decreases LPS-induced systemic inflammation in mice and the signal molecules involved remain unclear. These were confirmed by a semi-quantitative analysis of Iba-1 immunoreactivity and PAS staining in liver, kidney, and lung,respectively (Figures 4d, e, and f). To examine whether WT PDZ peptide can alter LPS-induced tissue damage in the kidney, cell toxicity assay was performed (Figure 3g). LPS induced cell damage in the kidney, however, WT PDZ peptide could significantly alleviate the toxicity, but mut PDZ peptide could not. Because cytotoxicity caused by LPS is frequently due to ROS production in the kidney (Su et al., 2023; Qiongyue et al., 2022), ROS production in the mitochondria was investigated in renal mitochondria cells harvested from kidney tissue (Figure 3h)......

Figure legend section: Indicated scale bars were 20 μm. (d,e,f) Semi-quantitative analysis of each are positive for Iba-1 in liver and kidney, and positive cells of PAS in lung, respectively. (g) After the kidneys were harvested, tissue lysates were used for MTT assay. (h) After.....

(4) In Figure 6G, H, the authors confirmed the change in expression of the M2 markers by PDZ peptide using the mouse monocyte cell line Raw264.7. It would be good to add an experiment on changes in M1 and M2 markers caused by PDZ peptides in human monocyte cells (for example, THP-1).

We thank you for your comments. To determine whether PDZ peptide regulates M1/M2 polarization in human monocytes, we examined changes in M1 and M2 gene expression in THP-1 cells. As a result, wild-type PDZ significantly suppressed the expression of M1 marker genes (hlL-1β, hIL-6, hIL-8, hTNF-ɑ), while increasing the expression of M2 marker genes (hlL-4, hIL-10, hMRC-1). However, mutant PDZ did not affect M1/M2 polarization. These results suggest that PDZ peptide can suppress inflammation by regulating M1/M2 polarization of human monocyte cells. These results are for the reviewer's reference only and will not be included in the main content.

Minor point:The use of language is appropriate, with good writing skills. Nevertheless, a thorough proofread would eliminate small mistakes such as:• line 254, " mut PDZ+LPS/LPS (45.75%) " → " mut PDZ+LPS/LPS (47.75%) "• line 296, " Figure 6f " → " Figure 6h "

We changed these points into the manuscript.